# A Novel Subtype of Bovine Hepacivirus Identified in Ticks Reveals the Genetic Diversity and Evolution of Bovine Hepacivirus

**DOI:** 10.3390/v13112206

**Published:** 2021-11-02

**Authors:** Jian-Wei Shao, Luan-Ying Guo, Yao-Xian Yuan, Jun Ma, Ji-Ming Chen, Quan Liu

**Affiliations:** School of Life Sciences and Engineering, Foshan University, Foshan 528225, China; jwshao1988@163.com (J.-W.S.); guoluanyingfs@163.com (L.-Y.G.); yyxyaoxian@sina.com (Y.-X.Y.); majun@fosu.edu.cn (J.M.); jmchen@fosu.edu.cn (J.-M.C.)

**Keywords:** bovine hepacivirus, novel subtype, genetic diversity, evolution, tick, China

## Abstract

Hepaciviruses represent a group of viruses that pose a significant threat to the health of humans and animals. New members of the genus *Hepacivirus* in the family *Flaviviridae* have recently been identified in a wide variety of host species worldwide. Similar to the Hepatitis C virus (HCV), bovine hepacivirus (BovHepV) is hepatotropic and causes acute or persistent infections in cattle. BovHepVs are distributed worldwide and classified into two genotypes with seven subtypes in genotype 1. In this study, three BovHepV strains were identified in the samples of ticks sucking blood on cattle in the Guangdong province of China, through unbiased high-throughput sequencing. Genetic analysis revealed the polyprotein-coding gene of these viral sequences herein shared 67.7–84.8% nt identity and 76.1–95.6% aa identity with other BovHepVs identified worldwide. As per the demarcation criteria adopted for the genotyping and subtyping of HCV, these three BovHepV strains belonged to a novel subtype within the genotype 1. Additionally, purifying selection was the dominant evolutionary pressure acting on the genomes of BovHepV, and genetic recombination was not common among BovHepVs. These results expand the knowledge about the genetic diversity and evolution of BovHepV distributed globally, and also indicate genetically divergent BovHepV strains were co-circulating in cattle populations in China.

## 1. Introduction

The genus *Hepacivirus*, belonging to the family *Flaviviridae*, comprises a genetically diverse group of human and animal pathogens. The genome of hepaciviruses is an unsegmented, single-stranded, and positive-sense RNA molecule about 10 kb in length, which contains a 5′ untranslated region (UTR) and a 3′ UTR, and a single long open reading frame (ORF) encoding a single polyprotein. This polyprotein is further cleaved by cellular and viral proteases into three structural proteins (Core, E1, and E2) and seven nonstructural proteins (p7, NS2, NS3, NS4A, NS4B, NS5A, and NS5B) [1,2]. The Hepatitis C virus (HCV), the prototypical member of the genus *Hepacivirus*, is one of the leading causes of acute and chronic hepatitis, liver failure, and hepatocellular carcinoma in humans, and the infection rate of HCV worldwide is approximately 3%, with an estimated 58 million people suffering from chronic HCV infection (https://www.who.int/en/news-room/fact-sheets/detail/hepatitis-c, accessed on 27 July 2021). HCV exhibits a restricted host range pattern, and humans are the only natural host, although experimental infection of HCV in chimpanzees is possible [3]. Since 2011, some HCV-like viruses have been identified from a wide variety of mammalian hosts, including dogs [4], horses [5], monkeys [6], bats [7], rodents [8,9], cattle [10,11], and donkeys [12]. Some HCV-like viruses have been assigned to additional *Hepacivirus* species, and designated as *Hepacivirus A*–*N* based on their phylogenetic relationships and host range [13]. HCV-like viruses have been described in non-mammalian hosts, such as catshark [14], birds [15], fish and other vertebrates [16,17,18]. Moreover, some HCV-like viruses have also been detected in non-vertebrate hosts, specifically in mosquitos and ticks, although their true hosts are uncertain [19,20]. In addition, a novel HCV-like, provisionally named as *Hepacivirus P*, was identified in long-tailed ground squirrels in China [21].

Bovine hepacivirus (BovHepV), the only member of the species *Hepacivirus N*, is a newly confirmed member of the genus *Hepacivirus* and likely only infects cattle [22]. BovHepV is hepatotropic, and causes acute or persistent infections in cattle [10,23]. Since its first discovery in cattle from Germany [10] and Ghana [11] in 2015, it has been detected in Brazil [24,25], Turkey [26], USA [27], China [28,29,30,31], and Italy [32], suggesting a worldwide distribution of BovHepV. As per the demarcation criteria for HCV genotyping and subtyping, BovHepV strains have been classified into two genotypes, and genotype 1 could be further divided into seven subtypes (A to G) [28,31]. In China, BovHepV has been detected in cattle in Guangdong, Jiangsu, Yunnan, and Sichuan provinces, with the positive rate of viral RNA ranging from 2.78% to 7.84% [28,29,30,31]. All of these BovHepV strains identified in China belong to genotype 1, with the exception of a strain in the genotype 2 [31].

In this study, three novel BovHepV strains were identified in blood-sucking ticks collected from cattle in Guangdong, China by using the unbiased high-throughput sequencing, which extends the knowledge about the genetic diversity and evolution of BovHepV, and indicates that genetically divergent BovHepV strains were co-circulating in cattle populations in China.

## 2. Materials and Methods

### 2.1. Tick Samples Collection

From June to July in 2020, 300 blood-sucking adult ticks were collected from cattle in Zhanjiang, Guangdong, Southern China. The tick species were identified following morphological criteria and further confirmed by sequencing and analyzing the 16S ribosomal RNA (*rrs*) gene of ticks [33]. Ticks were pooled, ten ticks per pool, and stored at −80 °C for further use.

### 2.2. RNA Extraction and Meta-Transcriptome Sequencing

Each pooled tick sample was soaked in 70% ethanol for 30 min, and washed with double distilled water three times. The samples were homogenized in 500 μL sterile phosphate-buffered saline (PBS), and their total RNA was extracted from 200 μL homogenates using the TRIzol LS reagent (Invitrogen, Carlsbad, CA, USA) and subsequently purified using the RNeasy Plus Mini Kit (Qiagen, Hilden, Germany). The quantity and quality of extracted RNA was evaluated with a NanoDrop 2000 (Thermo Fisher Scientific, Waltham, MA, USA). The extracted RNA was aliquoted and stored at −80 °C. One aliquot of each extracted RNA was then merged as two pools in equal quantity, and the quality of the pooled RNA was evaluated using an Agilent 2100 Bioanalyzer (Agilent Technologies, Santa Clara, CA, USA) before library construction and sequencing.

For library preparation, ribosomal RNA (rRNA) in the pooled RNA was removed using a Ribo-Zero-Gold (Epidemiology) kit (Illumina Inc., San Diego, CA, USA) following the manufacturer’s instructions, and the remaining RNA was fragmented, reverse-transcribed, adaptored, purified, and examined by the Agilent 2100 Bioanalyzer and ABI StepOnePlus Real-Time PCR System. Paired-end (150-bp) sequencing was performed on the Illumina Hiseq2500 platform. All library preparation and sequencing were performed by Novogene (Tianjin, China).

### 2.3. Bioinformatics Analyses and Genome Sequence Determination

Sequencing reads were adaptor- and quality-trimmed using the FASTP program [34] before de novo assembly using the Megahit program [35], with default parameter settings. The assembled contigs were compared against the database comprising all reference virus proteins using the diamond BLASTX program [36] with an e-value threshold of 1e–4, and the putative viral contigs were further compared to the non-redundant nucleotide and protein database to eliminate false positives. The confirmed viral contigs with unassembled overlaps or from the same scaffold were merged using the SeqMan program implemented in the Lasergene software package (version 7.1, DNAstar, Madison, WI, USA). To verify the assembly results, reads were mapped back to the target contigs with Bowtie2 [37], and inspected using integrated genomics viewer (IGV) [38] for any assembly errors. Gaps between these contigs were filled by RT-PCR and Sanger sequencing. The genome terminal of virus was determined by using 5′/3′ RACE kits (TaKaRa, Dalian, China) as described previously [39]. The complete viral genome was confirmed by Sanger sequencing with overlapping primers that covered the entire genome, which were designed based on the assembled sequences.

### 2.4. Sequence Comparison and Phylogenetic Analyses

The prediction of potential open reading frames (ORFs) was performed by ORFfinder (https://ftp.ncbi.nlm.nih.gov/genomes/TOOLS/ORFfinder/linux-i64/, accessed on 4 May 2021), and annotated based on comparisons against the non-redundant protein database. The cleavage sites for the viral polyprotein processing were extrapolated by manually comparing the polyprotein sequence with previously described BovHepV B1 strain identified in Germany [10]. N-glycosylation sites were predicted using NetNGlyc 1.0 (http://www.cbs.dtu.dk/services/NetNGlyc, accessed on 18 March 2012). The sequences of viruses were retrieved from the GenBank database, and sequences identities of nucleotide (nt) and amino acid (aa) were calculated by MegAlign program available within the Lasergene software package (version 7.1, DNAstar).

The phylogenetic relationships were estimated using the maximum-likelihood method (ML) implemented in PhyML version 3.0 [40], employing the general time reversible (GTR) nucleotide substitution model with a gamma (Γ)-distribution model of among-site rate variation and a proportion of invariable sites (i.e., GTR + Γ + I) and a subtree pruning and regrafting (SPR) branch-swapping algorithm. The support values on the phylogenetic trees were calculated from 100 bootstrap replicate.

### 2.5. Recombination Analyses

To find potential recombination events involved in the evolutionary history of BovHepV, we used the RDP, GENECONV, bootscan, maximum chi-square, Chimera, SISCAN, and distance plot methods available within RDP4 program [41]. The analyses were performed with default settings for the different test methods and a Bonferroni corrected *p* value cutoff of 0.05, and only sequences with significant evidence (*p* < 0.05) of recombination, namely, (i) detected by at least two methods and (ii) confirmed by phylogenetic analysis, were considered to represent strong evidence for recombination. Additionally, sequence alignment was also analyzed using similarity plot and bootscan analysis methods as implemented in Simplot version 3.5.1 [42].

### 2.6. Selection Pressures Analyses

The numbers of synonymous nt substitutions per synonymous site (d*S*) and the numbers of nonsynonymous substitutions per nonsynonymous site (d*N*) for each coding region of hepaciviruses identified in bovine, humans, and equine were calculated using the Nei-Gojobori model [43] implemented in MEGA version 7.0 [44]. To assess the selection pressure involved in the evolution of the hepacivirus genome, the site-specific selection pressures were estimated across the sequence of the entire polyprotein-coding gene using single-likelihood ancestor counting (SLAC) method as implemented in the Datamonkey web server (http://www.datamonkey.org, accessed 29 July 2010), and plotting the difference between d*N* and d*S* rates (d*N* − d*S*) for each codon. Additionally, the codon-specific analyses of the entire polyprotein-coding region were also assessed using three methods performed in the Datamonkey web server: fixed effects likelihood (FEL), fast unconstrained Bayesian approximation (FUBAR), and mixed effects model of evolution (MEME). Only codons with significance of *p* < 0.05 or a posterior probability > 0.95 identified by at least three methods were considered to be subject to positive selection [45].

## 3. Results

### 3.1. Identification of Three BovHepV in Ticks

A total of 300 adult ticks, which were identified as *Rhipicephalus microplus*, were collected from cattle in Zhanjiang city, Guangdong province, from June to July in 2020. After default quality control (QC) and de-barcoding steps, a total of 10,627,704 and 7,924,932 paired-end clean reads were generated in the libraries of GDZJ-02 and GDZJ-0103, respectively. Through de novo assembly and compared against nr database using diamond BLASTX program, 10 contigs with 388 to 1112 nt in length were annotated as sequences of BovHepV, with 92.5% to 98.2% amino acid sequence identity.

We then identified BovHepV from the 30 original RNA samples (each was pooled from 10 ticks) using RT-PCR. The complete genomic sequences of the BovHepV in these three samples were determined using RT-PCR and the RACE method, and were designated as BovHepV/GDZJ02, BovHepV/GDZJ02-2, and BovHepV/GDZJ02-3. These sequences have been deposited in GenBank under accession numbers MZ221927, MZ540979, and MZ540980, respectively. Using the complete genome sequence of BovHepV/GDZJ02 as the reference sequence, 224 reads were remapped to this reference sequence and provided 92.1% genome coverage (8115 nt/8808 nt) with 99.7% pairwise identity at a mean depth of 3.8×, and the percentage virus reads of the total number of nonribosomal reads is 0.0038% (224 reads/5,883,446 reads) (Figure 1).

### 3.2. Genomic Features of the Newly Identified BovHepV

The genomes of the above three BovHepV strains consisted of 8808 nt, with the same G + C contents of 52.7%, which was similar to those of other BovHepV strains (51.5–53.8%). These three genomes contain a large ORF with 8340 nt in size, flanked by 5′ UTR (263 nt) and 3′ UTR (205 nt). The large ORF encoded a predicted polyprotein of 2779 aa, which was further cleaved into 10 typical hepacivirus proteins in the order of Core–E1–E2–p7–NS2–NS3–NS4A–NS4B–NS5A–NS5B (Figure 2A). The putative cleavage sites specific for the processing of polyprotein were shown to be well conserved among the BovHepV strains (Table 1). Similar to HCVs and other hepaciviruses, two and seven N-glycosylation sites were also predicted in the pupative E1 and E2 proteins, respectively (Figure 2A).

### 3.3. Sequences Comparison of BovHepV

The complete polyprotein-coding gene of these three virus strains identified herein are highly homogenous to each other, with 99.6−99.8% nt identity and 99.6−99.8% aa identity (Table 2). In addition, they shared 67.7−84.8% nt identity and 76.1–95.6% aa identity to other BovHepV strains identified worldwide, while sharing the highest identities (84.4−84.8% nt identity and 94.8−95.6% aa identity) with the strains identified from Brazil (accession numbers MG781019 and MG781018). Meanwhile, these three newly identified strains shared 67.7−81.7% nt identity and 76.1−94.6% aa identity with other strains identified in China (Table 2). According to the cut-off values for HCV subtyping that the nucleotide sequence identity < 85% [46], all known BovHepV strains globally can be classified into two genotypes, and genotype 1 has strains identified worldwide, while genotype 2 was only identified in China and Brazil. BovHepV in different genotypes showed 75.9−76.8% aa identity and BovHepV strains in the same genotypes showed 91.1−100.0% aa identity (Table 2).

### 3.4. Phylogenetic Relationships of BovHepV

Phylogenetic relationships inferred based on the sequences of the complete polyprotein, NS3 and NS5 coding regions all suggested that BovHepV strains can be classified into two genotypes, and genotype 1 strains were clearly segregated into eight well-separated subtypes (A−H) including the novel one (subtype H) corresponding to the three BovHepV strains identified in this study, which was supported by >90% bootstrap value at the key nodes (Figure 3). The three viruses identified in this study presented the closest relationship with that identified in Brazil (subtype D), but divergent from other sequences identified in China, suggesting the presence of multiple subtypes in China. Similar patterns were observed for Germany, Ghana, and Brazil (Figure 3A–C). Remarkably, all sequences of BovHepV strains identified in China were divided into two genotypes and three subtypes in genotype 1 also indicate the high genetic diversity of BovHepV strains circulating in China.

### 3.5. Recombination and Mutation Analysis of BovHepV

No statistically supported recombination event was detected within BovHepV strains after systematic analyses. All subtypes of genotype 1 group clearly showed the high degree of similarity with each other (Figure 2B). Among the genotype 1 strains, the sequence of subtype H exhibited more than 90% amino acid sequence identity within the nearly complete polyprotein sequences of subtypes A−G, except for a small decrease in the partial NS5A protein of BovHepV subtype F. In addition, the sequences of subtype H showed high amino acid sequence divergence with genotype 2 strains in core, E1, E2, p7, partial NS2 and NS5B proteins, that shared < 90% amino acid sequence identity.

No frameshift mutations were identified in the genomes of HCV, BovHepV, and equine hepaciviruses (EqHV). In contrast, nucleotide substitution and insertion or deletion likely dominate the evolution of these hepaciviruses (Appendix A).

### 3.6. Selection Pressures on the Hepacivirus Genome

The d*N* and d*S* values of each site in the genome sequences of HCV, BovHepV, and EqHepV strains were calculated. Negatively selected sites (d*N* − d*S* < 0) were observed predominantly across the whole polyprotein-coding region of hepaciviruses (Figure 4), and the mean d*N* − d*S* of hepaciviruses identified in bovine, human, and equine were −1.55, −0.90, and −1.98, respectively. Meanwhile, all d*N*/d*S* ratio estimated in the coding sequences of ten proteins of hepaciviruses were lower than 1 (Table 3). Additionally, 1, 1, and 3 putatively positive selection sites in the polyprotein coding region of bovine, human, and equine hepaciviruses were predicted by three methods (FEL, FUBAR, and MEME) (Table 4).

## 4. Discussion

As a worldwide distributed human pathogen, HCV has posed a great threat to human health that causes liver failure, hepatitis, and hepatocellular carcinoma. Likewise, BovHepV causes acute or persistent infections in cattle [10,23]. BovHepV infections in cattle have been identified in five continents, and two genotypes and several subtypes of BoVHepV have been described (Figure 3), suggesting the ubiquitous presence of this highly variable virus. In this study, the complete genomes of three novel BovHepV strains were determined and parsed. Sequences comparison and phylogenetic analysis suggested that these three BovHepV strains constituted a novel subtype group of BovHepV, and hence expanded the known diversity of BovHepV.

Previous studies based on the limited BovHepV sequences provided evidence that subtypes of BovHepV are associated with their geographic origins [10,11,25,26]. However, this study demonstrated that two or more genotypes or subtypes co-circulated in countries such as China, Germany, Ghana, and Brazil (Figure 3), suggesting a complex geographic distribution of BovHepV genotypes and subtypes worldwide. For example, this study suggested that at least four subtypes (three in genotype 1 and one in genotype 2) co-circulate in China. This could have resulted from frequent international trade of live cattle, which can facilitate transboundary transmission of BovHepV. In addition, BovHepV has only been detected in cattle from limited areas of China, i.e., Guangdong, Jiangsu, Yunnan, and Sichuan provinces [28,29,30,31]. Epidemiological surveys in broader areas may better reveal the genetic diversity and epidemiological characteristics of BovHepV circulating in China.

In the present study, both d*N*/d*S* ratio analyses in the coding sequences of individual proteins, and the site-specific selection pressures analyses across the entire polyprotein-coding sequence confirmed the predominance of purifying selection in the genomic evolution of bovine hepaciviruses, which was consistent with previous studies conducted in other hepaciviruses [12,47,48], collectively suggesting that purifying selection is the dominant evolutionary pressure acting on the hepaciviruses genome. Interestingly, 1, 1, and 3 putative positively selected codons in the polyprotein coding region of bovine, human, and equine hepaciviruses were simultaneously predicted by FEL, FUBAR, and MEME methods. However, the functional significance of these sites putatively under positive selection should be experimentally assessed.

In this study, BovHepV strains were detected in blood-sucking ticks collected in cattle. However, as per the epidemiology of hepaciviruses, ticks are more likely to be the mechanical carrier rather than the arthropod vector for this virus, although ticks serve as the hosts of a variety of other viruses [13]. The low abundance of BovHepV in ticks in this study, and the similar situation of hepacivirus in Australian ticks [20], suggest that hepaciviruses identified in ticks were more likely to derive from the tick’s vertebrate host and were present in the blood contained in the engorged tick rather than being from the tick itself, which need further verification.

In conclusion, a new BovHepV subtype was identified in ticks in Guangdong, China through unbiased RNA sequencing, which expands the knowledge about the genetic diversity and evolution of BovHepV, and shows extensive genetically divergent BovHepV strains in China.

## Figures and Tables

**Figure 1 viruses-13-02206-f001:**
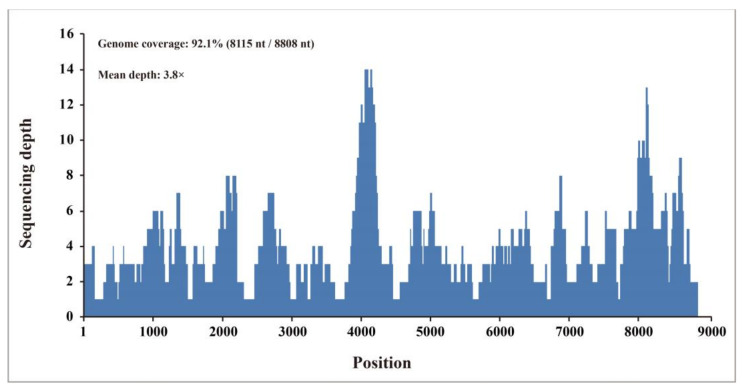
Mapped read count plot of the *Hepacivirus N* isolate GDZJ02 genome. The histograms show the coverage depth per base of the *Hepacivirus N* isolate GDZJ02 genome. The mean sequencing depth of the *Hepacivirus N* isolate GDZJ02 genome was 3.8×.

**Figure 2 viruses-13-02206-f002:**
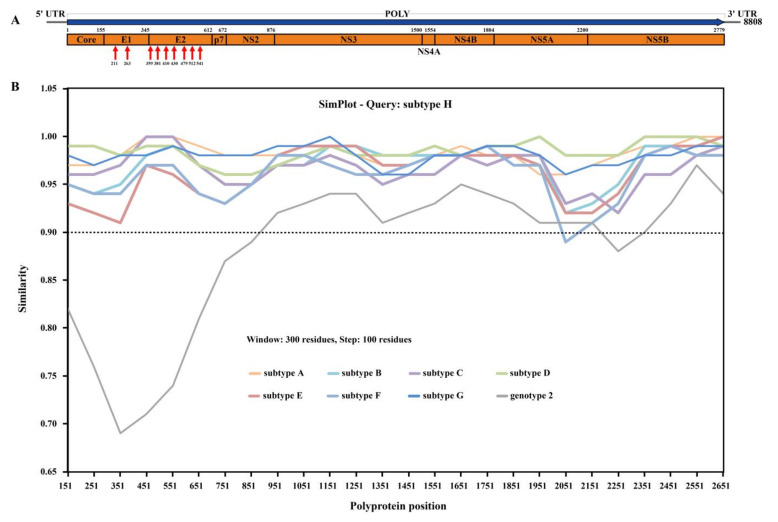
Genomic characterization of bovine hepacivirus. (**A**) Genome organization of BovHepV identified in this study. Red arrows indicate N-linked glycosylation sites. (**B**) Simplot analysis of BovHepV strains of subtype A-H based on polyprotein. Comparison of amino acid sequence identity of polyprotein within subtype A-H of BovHepV strains, calculated using Simplot version 3.5.1 with a sliding window of 300 and a step size of 100 residues. Different colors represent different subtypes of BovHepV. The GenBank accession numbers of the BovHepV strains of subtype A-H are listed in Table 1.

**Figure 3 viruses-13-02206-f003:**
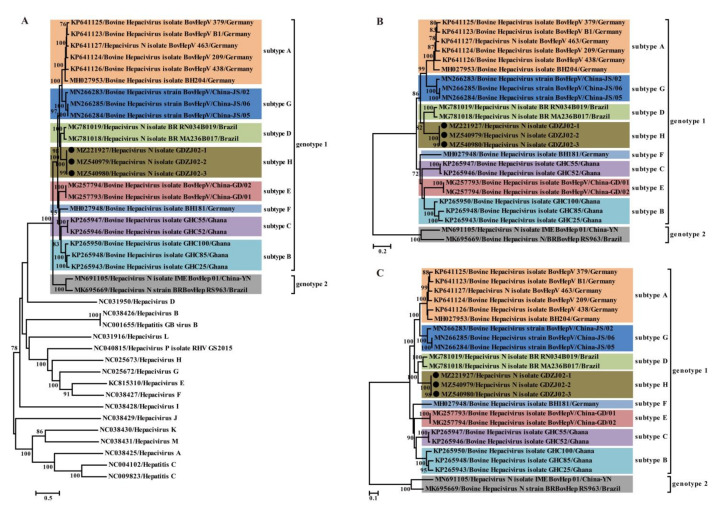
Phylogenetic analysis based on the nucleotide sequences of complete polyprotein-coding region (**A**), NS3 (**B**) and NS5 (**C**) of hepaciviruses including the newly identified sequences and other reference sequences retrieved from GenBank. The trees were constructed based on the maximum likelihood method implemented in PhyML v3.0, and mid-point rooted for clarity and the scale bar represents the number of nucleotide substitutions per site. Bootstrap values were calculated with 100 replicates of the alignment, and only bootstrap values > 70% are shown at relevant nodes. GenBank accession numbers are followed by the name of hepacivirus strains. Black dots indicate the BovHepV determined in this study.

**Figure 4 viruses-13-02206-f004:**
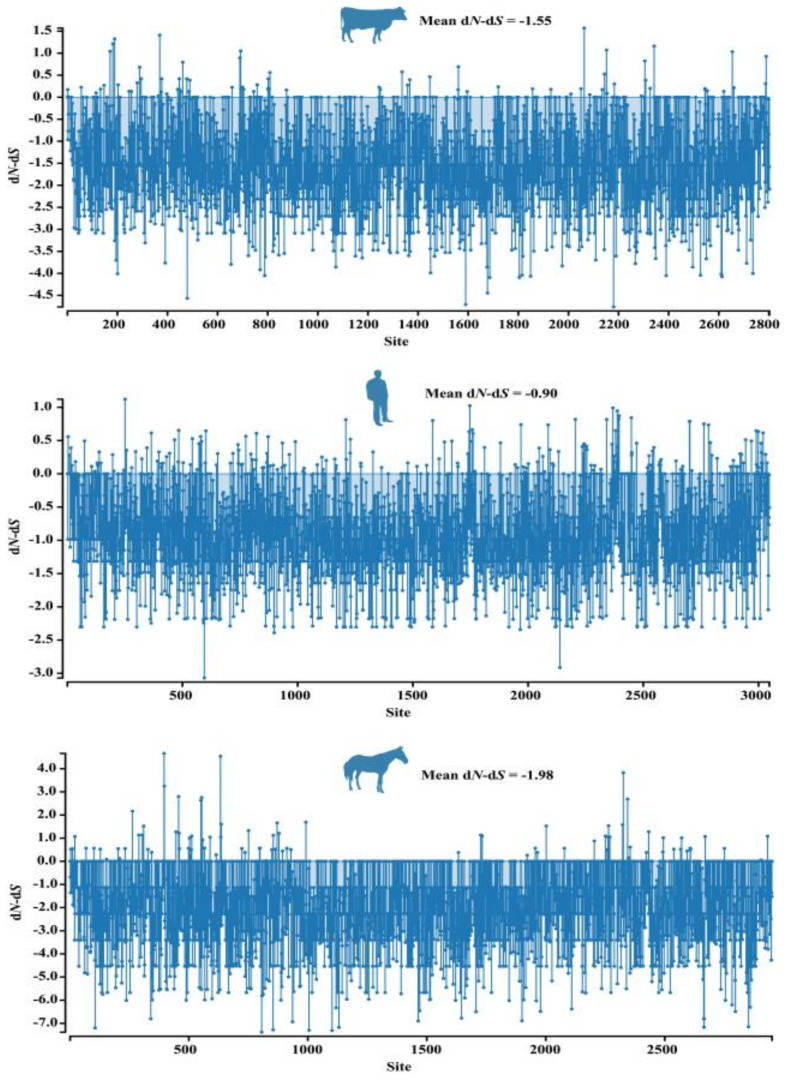
Differences between non-synonymous and synonymous (d*N* − d*S*) rates plotted for the complete polyprotein-coding region of hepaciviruses identified in bovine, humans, and equine. d*N* − d*S* < 0 indicates a negatively selected site, and d*N* − d*S* > 0 indicates a positively selected site.

**Table 1 viruses-13-02206-t001:** Comparison of predicted BovHepV polyprotein cleavage sites.

Virus	Cleavage Site at:								
	C/E1	E1/E2	E2/p7	P7/NS2	NS2/NS3	NS3/NS4A	NS4A/NS4B	NS4B/NS5A	NS5A/NS5B
1 KP641123/Germany	VSG/YRH	VEA/TTT	ATA/ALL	VTA/LDF	APC/SPI	LDV/WGA	EEC/WGF	VPC/GFN	KEC/SYS
2 KP641125/Germany	…/..Q	…/…	…/…	…/..S	…/…	…/…	…/..L	…/…	…/…
3 KP641127/Germany	…/..Q	…/…	…/…	…/…	…/…	…/…	…/…	…/…	…/…
4 KP641124/Germany	…/..Q	…/…	…/…	…/…	T../…	…/…	…/..L	…/…	…/…
5 KP641126/Germany	…/…	…/…	…/…	…/…	…/…	…/…	…/..L	…/…	…/…
6 MH027953/Germany	…/..Q	…/…	…/…	…/…	…/…	…/…	…/..L	…/…	…/…
7 KP265950/Ghana	…/..L	…/…	…/…	…/..I	…/A..	…/…	…/..L	…/…	…/…
8 KP265948/Ghana	…/..M	…/…	…/…	…/..V	…/A..	…/…	…/..L	…/…	…/…
9 KP265943/Ghana	…/..L	…/…	…/…	…/..V	…/A..	…/…	…/..L	…/…	…/…
10 KP265947/Ghana11 KP265946/Ghana	.D./…	…/…	…/…	…/..V	…/…	…/…	…/..L	.Q./…	…/…
	.D./…	…/…	…/…	…/..V	…/…	…/…	…/..L	.Q./…	…/…
12 MG781019/Brazil	.D./..Q	…/…	…/…	…/..S	…/…	…/…	…/..L	…/…	…/…
13 MG781018/Brazil	.D./..Q	…/…	…/…	…/..S	…/…	…/…	…/..L	…/…	.K./…
14 MG257793/China-GD	.D./..L	…/…	.S./…	…/…	…/…	…/…	…/..L	…/…	…/…
15 MG257794/China-GD	.D./..L	…/…	.S./…	…/…	…/…	…/…	…/..L	…/…	…/…
16 MH027948/Germany	.D./..L	…/…	…/…	…/…	…/…	…/…	…/..L	…/…	…/…
17 MN266283/China-JS	…/..Q	…/…	…/…	…/..L	…/…	…/…	…/…	…/…	…/…
18 MN266285/China-JS	…/..Q	…/…	…/…	…/.NL	…/…	…/…	…/..L	…/…	…/…
19 MN266284/China-JS	…/..Q	…/…	…/…	…/.NL	…/…	…/…	…/..L	…/…	…/…
20 MZ221927/China-GD	.H./..Q	…/…	…/…	…/..S	…/…	…/…	…/..L	…/…	…/…
21 MZ540979/China-GD	.H./..Q	…/…	…/…	…/..S	…/…	…/…	…/..L	…/…	…/…
22 MZ540980/China-GD	.H./..Q	…/…	…/…	…/..S	…/…	…/…	…/..L	…/…	…/…
23 MN691105/China-YN	TDA/..Y	..T/..E	…/…	…/D..	V../…	…/…	…/…	…/…	R../…

Cleavage is indicated by a slash (/). The sequences shaded with different colors indicate different subtype of BovHepV. Subtype A (
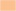
), subtype B (
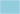
), subtype C (
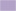
), subtype D (
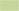
), subtype E (
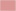
), subtype F (
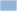
), subtype G (
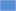
), subtype H (
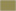
), and genotype 2 (
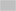
).

**Table 2 viruses-13-02206-t002:** The sequence identity within the BovHepV polyprotein gene at the nucleotide (upper right) and amino acid (lower left, boldface) levels calculated using ClustalW method implemented in MegAlign. The compared strains were indicated as accession number and country, and the different colors mean the different subtypes/genotypes of the BovHepV strains.

Strains	1	2	3	4	5	6	7	8	9	10	11	12	13	14	15	16	17	18	19	20	21	22	23
1 KP641123/Germany		93.8	93.6	93.3	91.0	90.8	80.2	79.9	80.0	80.5	80.4	82.7	82.5	79.7	79.6	80.2	84.5	84.6	84.6	82.0	82.0	81.9	66.8
2 KP641125/Germany	**98.2**		93.7	93.2	91.5	91.1	80.3	79.9	80.2	80.6	80.4	82.9	82.6	79.6	79.6	80.2	84.4	84.1	84.1	81.8	81.7	81.6	66.7
3 KP641127/Germany	**97.9**	**98.0**		93.5	91.8	91.0	80.1	80.1	80.4	80.7	80.5	83.0	82.6	79.7	79.7	80.4	84.5	84.6	84.6	82.0	81.9	81.8	66.9
4 KP641124/Germany	**98.1**	**97.9**	**98.0**		91.6	91.0	80.0	79.9	80.1	80.8	80.6	82.6	82.6	79.6	79.6	80.2	84.6	84.5	84.6	81.9	81.8	81.7	67.3
5 KP641126/Germany	**97.4**	**97.5**	**97.4**	**97.5**		90.6	79.8	79.8	80.2	80.4	80.3	82.4	82.3	79.0	79.1	79.7	84.4	84.0	84.0	81.9	81.7	81.7	66.9
6 MH027953/Germany	**97.0**	**97.1**	**97.4**	**97.2**	**96.9**		79.6	79.8	79.9	80.2	79.9	82.6	82.4	79.7	79.7	79.8	84.1	83.9	84.0	81.8	81.7	81.6	66.7
7 KP265950/Ghana	**93.0**	**92.6**	**92.8**	**92.9**	**92.5**	**92.5**		90.4	90.9	82.9	82.8	80.6	80.8	82.1	82.1	81.7	80.3	80.1	80.1	80.2	80.1	80.0	67.3
8 KP265948/Ghana	**93.2**	**92.9**	**93.0**	**93.1**	**92.6**	**92.7**	**98.7**		92.1	82.8	82.5	80.6	80.3	81.8	81.8	81.2	80.1	79.9	79.9	79.9	79.8	79.7	67.6
9 KP265943/Ghana	**92.9**	**92.8**	**92.9**	**92.8**	**92.6**	**92.7**	**98.6**	**99.0**		83.0	82.8	80.8	80.5	81.9	81.8	81.5	80.4	80.0	80.1	80.2	80.0	80.0	67.4
10 KP265947/Ghana	**93.0**	**92.8 **	**92.9**	**93.0**	**92.8 **	**92.4**	**96.2**	**96.4**	**96.3**		98.4	81.2	81.2	81.7	81.7	81.6	80.1	80.2	80.2	80.0	79.8	79.8	66.9
11 KP265946/Ghana	**92.4**	**92.1**	**92.1**	**92.2**	**92.1**	**91.7**	**95.5**	**95.6**	**95.6**	**98.9**		81.0	81.0	81.6	81.5	81.5	79.9	80.0	80.0	79.8	79.7	79.6	66.9
12 MG781019/Brazil	**94.3**	**93.6**	**93.7**	**94.0**	**93.8**	**93.9**	**93.2**	**93.2**	**93.0**	**93.0**	**92.4**		93.9	80.8	80.8	80.1	82.5	82.5	82.5	84.8	84.6	84.6	67.1
13 MG781018/Brazil	**94.5**	**93.9 **	**93.9**	**94.1**	**93.9**	**94.2**	**93.2**	**93.1**	**93.0**	**93.2**	**92.6**	**98.1**		80.3	80.3	80.1	82.3	82.2	82.2	84.6	84.4	84.4	66.8
14 MG257793/China-GD	**92.0**	**91.9**	**91.9**	**91.8**	**91.4 **	**91.6**	**95.4**	**95.3 **	**95.3**	**94.8**	**94.4**	**92.3**	**92.4**		99.8	81.1	80.0	79.4	79.4	80.7	80.5	80.4	67.1
15 MG257794/China-GD	**92.0**	**91.9**	**91.9 **	**91.9**	**91.4**	**91.6**	**95.4**	**95.3**	**95.4**	**94.8**	**94.4**	**92.4**	**92.5**	**100**		81.1	79.9	79.4	79.4	80.6	80.5	80.4	67.1
16 MH027948/Germany	**92.1**	**91.9**	**92.1**	**92.0**	**91.9**	**91.8**	**95.1**	**94.9**	**94.9**	**95.0**	**94.4**	**92.7**	**92.5**	**94.3**	**94.3**		79.6	79.8	79.8	79.7	79.6	79.5	67.6
17 MN266283/China-JS	**95.7**	**95.4**	**95.7**	**95.6**	**95.6**	**95.6**	**93.1**	**93.2**	**93.0**	**93.0**	**92.5**	**94.8**	**94.7**	**92.3**	**92.3**	**92.3**		92.7	92.8	81.7	81.6	81.5	67.5
18 MN266285/China-JS	**95.5**	**95.3**	**95.6**	**95.5**	**95.4**	**95.4**	**92.6**	**92.8**	**92.6**	**92.7**	**92.2**	**94.4**	**94.5**	**92.0**	**92.0**	**92.0 **	**97.9**		99.9	81.4	81.2	81.2	67.2
19 MN266284/China-JS	**95.6**	**95.4**	**95.7**	**95.6**	**95.5**	**95.5**	**92.7**	**92.9**	**92.6**	**92.8**	**92.3**	**94.5**	**94.6**	**92.0**	**92.0**	**92.1**	**98.0**	**99.8**		81.4	81.3	81.2	67.2
20 MZ221927/China-GD	**94.4**	**94.1 **	**93.9**	**94.1**	**94.0**	**94.0**	**92.6 **	**92.7**	**92.6**	**92.6**	**92.0**	**95.5**	**95.6 **	**91.8**	**91.8**	**92.0**	**94.6**	**94.2**	**94.3**		99.7	99.6	66.8
21 MZ540979/China-GD	**94.0**	**93.8**	**93.5**	**93.7 **	**93.6**	**93.6**	**92.1**	**92.3 **	**92.2**	**92.1**	**91.6**	**95.0 **	**95.1**	**91.3**	**91.4**	**91.5**	**94.1**	**93.7**	**93.8**	**99.4**		99.8	66.8
22 MZ540980/China-GD	**93.8**	**93.6**	**93.2**	**93.4**	**93.4**	**93.4**	**91.9**	**92.0**	**92.0**	**91.9**	**91.4**	**94.8**	**94.9**	**91.1**	**91.1**	**91.3**	**93.9**	**93.5**	**93.6**	**99.1**	**99.7**		66.7
23 MN691105/China-YN	**76.5**	**76.4**	**76.2 **	**76.4**	**76.4 **	**75.9 **	**76.2 **	**76.4**	**76.4**	**76.8**	**76.6**	**76.7**	**76.6**	**75.9**	**75.9**	**76.6**	**76.7**	**76.4**	**76.4**	**76.5 **	**76.3**	**76.1**	

The indication of different colors same as that in Table 1.

**Table 3 viruses-13-02206-t003:** Comparison of the mean numbers of nonsynonymous (d*N*) and synonymous (d*S*) substitutions per site, and their ratios, in the coding regions of hepacivirus in bovine, humans, and equine.

Gene	Bovine Hepacivirus (*n* = 23)			Human Hepacivirus (*n* = 40)			Equine Hepacivirus (*n* = 11)		
	d*N*	d*S*	d*N*/d*S*	d*N*	d*S*	d*N*/d*S*	d*N*	d*S*	d*N*/d*S*
Core	0.081	0.968	0.084	0.061	0.648	0.094	0.029	0.600	0.048
E1	0.094	1.389	0.068	0.243	1.958	0.124	0.047	0.817	0.058
E2	0.089	1.307	0.068	0.174	2.161	0.081	0.036	1.027	0.035
p7	0.078	1.517	0.051	0.376	1.830	0.205	0.048	0.919	0.052
NS2	0.086	1.370	0.063	0.332	2.281	0.146	0.049	1.077	0.045
NS3	0.032	1.164	0.027	0.119	2.154	0.055	0.006	0.915	0.007
NS4A	0.060	1.275	0.047	0.169	2.140	0.079	0.018	1.269	0.014
NS4B	0.027	1.487	0.018	0.279	0.460	0.607	0.011	0.882	0.012
NS5A	0.074	1.188	0.062	0.245	2.227	0.110	0.033	0.589	0.056
NS5B	0.041	1.056	0.039	0.157	1.452	0.108	0.027	0.812	0.033

The GenBank accession number of viruses used in this analysis were shown in Appendix A.

**Table 4 viruses-13-02206-t004:** Prediction of putative positive selection site of hepacivirus genome from different hosts.

Model	Bovine Hepacivirus (*n* = 23)	Human Hepacivirus (*n* = 40)	Equine Hepacivirus (*n* = 11)
SLAC	0	0	0
FEL	**171**, 689, 2789	10, **75**, 444, 3018, 3026	**397, 633, 2326**
FUBAR	**171**, 182	**75**	**397**, 457, **633, 2326**
MEME	58, 108, 158, **171**, 279, 369, 434 444, 461, 497, 530, 586, 623, 658, 689, 692, 786, 1060, 1243, 1498, 1949, 1965, 2001, 2011, 2063, 2079, 2086, 2096, 2111, 2129, 2144, 2342, 2395, 2429, 2640, 2654, 2658, 2664, 2675, 2686, 2789, 2793	4, 10, 16, 20, 72, **75**, 191, 194, 195, 198, 200, 202, 211, 232, 257, 297, 357, 360, 444, 449, 466, 545, 561, 574, 683, 716, 775, 795, 830, 875, 1008, 1102, 1164, 1169, 1219, 1256, 1378, 1414, 1439, 1610, 1613, 1642, 1687, 1772, 1815, 1953, 1960, 1965, 1983, 1984, 1999, 2043, 2061, 2090, 2106, 2146, 2162, 2183, 2230, 2232, 2243, 2250, 2273, 2288, 2291, 2332, 2350, 2355, 2366, 2371, 2433, 2452, 2508, 2528, 2531, 2535, 2539, 2544, 2579, 2582, 2583, 2593, 2597, 2612, 2646, 2672, 2697, 2700, 2716, 2724, 2731, 2738, 2769, 2774, 2786, 2796, 2826, 2851, 2893, 2900, 2913, 2917, 2923, 2938, 2942, 2968, 3005, 3017, 3018, 3026, 3035, 3036, 3052	207, 263, 295, 305, 309, 388, **397**, 437, 502, 503, 554, 589, **633**, 634, 871, 894, 1004, 1189, 2057, 2204, **2326**, 2355, 2356, 2433, 2467, 2492, 2571, 2670

*p* < 0.05 or posterior probability > 0.95; positions identified as being under positive selection by at least three methods are shown in bold. The GenBank accession number of viruses used in this analysis were same as that in Appendix A.

## Data Availability

The sequences generated in this study have been submitted to GenBank under accession numbers MZ221927, MZ540979, and MZ540980. The information about the sequences used in the analysis of this study is shown in Appendix A.

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
