# Peer review of "A Novel Subtype of Bovine Hepacivirus Identified in Ticks Reveals the Genetic Diversity and Evolution of Bovine Hepacivirus"

_viruses, 2021, doi:10.3390/v13112206_

Round 1

Reviewer 1 Report

Manuscript Review: viruses-1430830-peer-review-v1

A novel subtype of bovine hepacivirus identified in ticks reveals the genetic diversity and evolution of bovine hepacivirus

Jian-Wei Shao, Luan-Ying Guo, Yao-Xian Yuan, Jun Ma, Ji-Ming Chen, Quan Liu

Summary:

The manuscript presented by Shao et al reports the detection of 3 novel bovine hepacivirus strains in ticks that had been feeding upon cattle in Guangdong, China. These novel viral sequences are described, and compared phylogenetically with other bovine hepacivirus genomes. Additionally, some statistical tests for purifying vs. diversifying selection were performed across human, bovine, and equine hepacivirus sequences.

Overall Opinion:

Overall, this article is quite well written and of interest in reporting 3 novel strains of bovine hepacivirus that likely correspond to a newly reported subtype of genotype 1. This manuscript is limited as it comprises an essentially descriptive study, however is of value to future work in the field. The analyses of purifying vs. diversifying selection do seem a little disjointed compared to the rest of the paper focus (rather than focusing upon the 3 novel strains; human, bovine, and equine hepaciviruses were considered as groups to determine sites putative selection pressures). The findings of potential positive selection are also undeniably speculative as they are not supported by subsequent experimental evidence, though in fairness the authors address this honestly.

I would consider this article worthy of publication. Though the findings are descriptive and potentially speculative, this manuscript will likely provide an important resource to inspire future work in the field.

Comments:

  1. Line 57 seems to have a mistake. The authors state that “BovHepV strains have been classified into one genotype”, but contradict this several times later in the article by mentioning 2 genotypes of this virus.

  1. Fig 2 – The annotation of putative glycosylation sites would be more valuable if the residue numbers were provided along with the arrows. Additionally, the residue numbers corresponding to each division of the polyprotein would be good to include in panel A.

  1. The discussion section does not address the DataMonkey analyses of site-specific selection at all. These results should be mentioned and discussed in context with the literature.

Author Response

Point 1: Line 57 seems to have a mistake. The authors state that “BovHepV strains have been classified into one genotype”, but contradict this several times later in the article by mentioning 2 genotypes of this virus.

Response: We have carefully checked and revised this mistake.

Point 2: Fig 2 – The annotation of putative glycosylation sites would be more valuable if the residue numbers were provided along with the arrows. Additionally, the residue numbers corresponding to each division of the polyprotein would be good to include in panel A.

Response: Revised accordingly in Figure 2.

Point 3: The discussion section does not address the DataMonkey analyses of site-specific selection at all. These results should be mentioned and discussed in context with the literature.

Response: Revised accordingly in the section of Discussion.

Reviewer 2 Report

The authors described the novel BovHepV identified in this study. Whole-genome sequences were determined and the result indicated these BovhepV would be classified into a novel subtype. Such kind of basic study will help to understand the epidemiology of BovHepV.

Minor Comments

Table 1 and 2: You have 3 MZ221927 in the table. Colors are difficult to recognize differences. I recommend describing subtypes in the table.

FIg. 2B: Lines of subtype A-G and genotype 2 should be colored.

Fig. 3: The same colors used in Tables 1 and 2 should be used. 

Figure legend is needed for Fig. S1.

Author Response

Point 1: Table 1 and 2: You have 3 MZ221927 in the table. Colours are difficult to recognize differences. I recommend describing subtypes in the table.

Response: We have carefully checked and revised the mistakes, and we have revised the table note to make the indication of different colours easier to understand in Table 1 and 2.

Point 2: Fig. 2B: Lines of subtype A-G and genotype 2 should be colored.

Response: Revised accordingly in Figure 2.

Point 3: Fig. 3: The same colors used in Tables 1 and 2 should be used.

Response: Revised accordingly in Figure 3.

Point 4: Figure legend is needed for Fig. S1.

Response: Revised accordingly, and it has been submitted in Supplementary files.
